# 2019 Survey of Antimicrobial Drug Use and Stewardship Practices in Adult Cows on California Dairies: Post Senate Bill 27

**DOI:** 10.3390/microorganisms9071507

**Published:** 2021-07-14

**Authors:** Essam M. Abdelfattah, Pius S. Ekong, Emmanuel Okello, Deniece R. Williams, Betsy M. Karle, Joan D. Rowe, Edith S. Marshall, Terry W. Lehenbauer, Sharif S. Aly

**Affiliations:** 1Veterinary Medicine Teaching and Research Center, School of Veterinary Medicine, University of California, Davis, Tulare, CA 93274, USA or essam.abdelfattah@fvtm.bu.edu.eg (E.M.A.); pekong@ucdavis.edu (P.S.E.); eokello@ucdavis.edu (E.O.); dvmwilliams@ucdavis.edu (D.R.W.); tlehenbauer@vmtrc.ucdavis.edu (T.W.L.); 2Department of Animal Hygiene and Veterinary Management, Faculty of Veterinary Medicine, Benha University, Moshtohor 13736, Egypt; 3Department of Population Health and Reproduction, School of Veterinary Medicine, University of California, Davis, CA 95616, USA; jdrowe@ucdavis.edu; 4Cooperative Extension, Division of Agriculture and Natural Resources, University of California, Orland, CA 95963, USA; bmkarle@ucanr.edu; 5Antimicrobial Use and Stewardship, Animal Health and Food Safety Services Division, California Department of Food and Agriculture, Sacramento, CA 95814, USA; edie.marshall@cdfa.ca.gov

**Keywords:** antimicrobial drug resistance, antimicrobial stewardship, California, dairy cattle, judicious use, survey

## Abstract

Antimicrobial resistance (AMR) is a global issue for both human and animal health. Antimicrobial drug (AMD) use in animals can contribute to the emergence of AMR. In January 2018, California (CA) implemented legislation (Senate Bill 27; SB 27) requiring veterinary prescriptions for medically important AMD use in food animals. The objective of our survey was to characterize AMD use, health management, and AMD stewardship practices of adult cows on CA dairies since the implementation of SB 27. In 2019, we mailed a questionnaire to 1282 California dairies. We received a total of 131 (10.2%) survey responses from 19 counties in CA. Our results showed that 45.6% of respondents included a veterinarian in their decision on which injectable AMD to purchase. Additionally, 48.8% of dairy producers included a veterinarian in their decision on which AMDs were used to treat sick cows. The majority (96.8%) of dairy producers were aware that all uses of medically important AMDs require a prescription. Approximately 49% of respondents agreed or strongly agreed that AMD use in livestock does not cause problems in humans. The survey documents antimicrobial use and stewardship practices in CA’s dairy industry and focus areas for future research and education.

## 1. Introduction

Antimicrobial drugs (AMDs) have been used to maintain or improve the health, productivity, and welfare of animals [1]. However, the use of AMDs increases the risk of antimicrobial resistance (AMR) [2], which is a global concern for both human and animal health. According to the 2019 U.S. Center for Disease Control and Prevention (CDC) report, more than 2.8 million AMR infections occur in human hospitals in the United States each year, and more than 35,000 people die as a result [3]. To control and prevent AMR, the U.S. Food and Drug Administration (FDA) published guidelines regulating therapeutic use of medically important antimicrobial drugs (MIADs) in feed and water for food-producing animals and prohibited the use of MIADs for production purposes, such as growth promotion and feed efficiency. In the United States, a MIAD is an antimicrobial that is important for treating human disease and includes all critically important, highly important, and important drugs listed in Appendix B of the federal Food and Drug Administration’s Guidance for Industry #152 (CVM GFI #152). The Veterinary Feed Directive (VFD) final rule issued by the FDA was implemented in January 2017 and mandated supervision by a licensed veterinarian for use of MIADs in feed or water under a valid veterinary client patient relationship (VCPR) [4]. In California, a VCPR is established when the client has authorized the licensed veterinarian to assume responsibility for making medical judgements and the need for medical treatment of the patient (including the prescription of AMDs) and the veterinarian has assumed that responsibility and has communicated with the client an appropriate course of treatment. For a valid VCPR, the veterinarian must be personally acquainted with the care of the animal(s) by hands-on examination of the animal or by medically appropriate and timely visits to the premises where the animals are kept and have enough knowledge of the animal(s) to give at least a general or preliminary diagnosis of the medical condition. CCR § 2032.1. California further supported this effort by passing the Use of Antimicrobial Drugs Law (California Food and Agricultural Code, FAC Sections 14400–14408) in 2015 [5], presented to the citizens of California for vote as Senate Bill (SB) 27, here onwards referred to as SB 27. The SB 27 regulations require veterinary oversight (prescription) for the use of all other dosage forms of MIADs in livestock, i.e., those products that were previously available over the counter (OTC), and were fully implemented in January 2018. California’s SB 27 also resulted in development of voluntary AMD stewardship guidelines, best management practices, and monitoring of AMD use. Similar legislation has been passed in Maryland, Maryland SB 471 [6], and several other states to increase AMD stewardship through judicious use of AMDs.

With the global efforts to promote AMD stewardship in animals and increased regulation of MIAD use in food animals, it is crucial to survey the dairy industry on the possible negative consequences that may arise because of attempts to alter AMD use [7]. Ancillotti et al. [8] found that informing the public about the consequences of inappropriate use of AMDs is not sufficient to induce behavioral changes and indicated that identifying the factors that promote and hinder the judicious use of AMDs is crucial for change of antimicrobial use. Additionally, emphasis on public and agricultural education is critical in promoting prudent AMD use in animal production [9]. A recent qualitative interview study conducted in New York [7] showed that conventional dairy producers had a low level of concern about the possible impacts of on-farm AMR on human health and believed that their AMD use was already judicious. In addition, a survey of beef producers in Tennessee [10] indicated that approximately 20% of the producers were either slightly familiar or not familiar with AMR. A survey of dairy producers in Michigan and Ohio [11] showed that 29% agreed that AMD use in agriculture makes it harder to treat future livestock infections, while only 7% agreed that AMD use in livestock leads to bacterial infections in people that are difficult to treat. Survey results in South Carolina [12] showed that most producers (86%) were not concerned that overuse of AMDs in animals could result in AMR among farm workers.

Exploring California dairy producers’ management changes related to AMD use and AMR, in light of SB 27 regulations, will help identify opportunities, barriers, and possible strategies to advance antimicrobial stewardship in California. The objectives of this survey were to (1) characterize AMD use and stewardship practices and health management of adult cows on CA dairies in 2019, after implementation of the VFD and SB 27; and (2) describe changes in AMD use and stewardship on dairies that responded to the current survey and its earlier mailing in 2018 (Ekong et al., 2021). We hypothesized that AMD use and stewardship and heath management will vary by region, herd demographics, and type of production (organic vs. conventional). A secondary hypothesis was that producers made changes in management and AMD stewardship practices between 2018 and 2019.

## 2. Materials and Methods

### 2.1. Study Design and Administration of Survey

A survey instrument was developed to collect information about antimicrobial use in adult cows in California dairies after implementation of SB 27. The current survey was adapted from a previous survey of CA dairies in 2018 [13], with only two new questions (Q28, and Q44E, Appendix B) added to the current one. The survey was reviewed by the Institutional Review Board of the University of California, Davis, and was granted exemption approval (IRB ID: 1537295-1; approved 9 January 2020).

The questionnaire consisted of 44 questions partitioned into three main sections (Appendix B): Section 1 (herd information); Section 2 (dairy cow health management and antimicrobial use); and Section 3 (practices and perspectives about AMDs and AMR). An optional section was included in the survey to allow respondents to provide contact information if they were interested in participating in a follow-up study and to provide feedback about the questionnaire. Multiple-point scales and ordinal Likert scales [14] were used to capture the participant responses to the survey questions. The questionnaire was pre-tested using in-person interviews with extension and outreach specialists and veterinarians in CA to identify questions that may have been confusing or ambiguous.

A list of all 1282 licensed grade A dairies in California in 2017 served as a sampling frame for our survey. Grade A dairies produce fluid milk that meets federal and local state regulations [15]. In the case of California, the legal limit for grade A milk bulk tank somatic cell count is 600,000 cells/mL. Each dairy identified on the list was randomly assigned a unique confidential number. Each dairy received a single questionnaire, and the survey responses from each participant represented the attributes of that unique farm.

A survey packet containing a copy of the questionnaire, a postage-paid addressed business reply envelope, and an information cover letter was mailed to each listed dairy. The survey questionnaire was made available to participants in print form with the option of completing it over the phone. The cover page introduced the dairy producers to the objective of the study and provided information about the term “antimicrobials”. Producers were instructed that, for the purpose of this survey, questions will refer to all antimicrobials as ‘antibiotics’ regardless of their origin. To improve the questionnaire response rate, each dairy was mailed a reminder card two weeks after the initial questionnaire package. A second copy of the questionnaire, followed by a reminder card, were similarly mailed three months later to dairies that had not previously responded. All mailings were completed between May and December 2019.

### 2.2. Questionnaire Sections

The 44 survey questions inquired about the respondent’s dairy herd demographics and their AMD use practices; factors driving their choice of antimicrobials; the impact of SB 27 on AMD purchases, usage, and costs; and their responses about AMDs and AMR in dairy cattle. The questionnaire consisted of three main sections, as follows.

Herd Information. The first section focused on gathering baseline information about the dairy producers, including CA county, type of production (conventional vs. organic), average herd size (number of milking cows), breeds, the herd’s annual rolling herd average (RHA) for milk production, and average bulk tank somatic cell count (BTSCC).

Dairy Cow Health Management and Antimicrobial Use. The second section addressed dry-off protocols, types of AMDs used at dry-off, disease prevention, and vaccinations used in adult cows. In addition, information was collected on AMD used to treat sick cows, who made the decisions on what AMDs are purchased and used to treat sick cows, and whether producers had written or computerized animal health records or used a drug inventory log on their dairies. Moreover, information was collected on how and which AMD treatment information is tracked, the farm’s disease diagnosis and management practices, and a description (written or verbal) of the VCPR on that farm.

Antimicrobial Stewardship Practices. The third section posed questions relating to the respondent’s participation in animal welfare audit programs and/or dairy quality assurance programs, the respondent’s familiarity with the FDA’s term MIADs, and awareness that all MIADs required a prescription and were no longer sold OTC in CA after 1 January 2018. Additional questions addressed differences that may have occurred in the periods before and after 1 January 2018 in the use of OTC and prescription AMDs on the dairy, changes made regarding previously available OTC AMDs, and changes in the dairy’s AMD drug costs. Similarly, questions inquired about differences before and after 1 January 2018 in the usage of alternatives to AMDs (defined as vaccines, vitamins, minerals, herbal remedies, or others), any changes in management to prevent a disease outbreak or disease spread, and changes noted in animal health on the dairy. The final questions in this section collected information about the level of agreement of producers with statements relating to AMD use, stewardship practices, and AMR in dairy cattle.

### 2.3. Statistical Analyses

Frequencies as well as proportions and their standard errors (SE) were computed for categorical and ordinal variables. Mean and SE were computed for continuous variables. Confidence intervals for proportions were calculated using the normal approximation method. Data on the location of dairies in CA were reclassified into three regions (Figure 1): Northern California (NCA), Northern San Joaquin Valley (NSJV), and Greater Southern California (GSCA) based on the distinct differences among the three regions in dairy infrastructure and management practices [16,17]. Herd size was categorized as <1305 milking cows (less than the state mean), 1305 to 3500 milking cows (larger herds), and >3500 milking cows (largest herds) based on a mean herd size of 1304 in the state of CA [18]. Rolling herd average values were converted to kilograms by dividing the number of pounds by 2.2046 (kg/cow) and reclassified into two categories, <10,880 kg/cow and ≥10,880 kg/cow based on CA state mean (10,880 kg/cow). For producers who reported their herd’s production in pounds/head/day, the estimate was multiplied by 305 to estimate the herd RHA, while BTSCC (cells/mL) was reclassified into three categories: <100,000, 100,000–199,999, and ≥200,000 based on the CA state mean BTSCC of 194,000 cells/mL [18]. For each disease condition, the treatment incidence rate was estimated for each respondent’s herd as cases per 100 milking cow months. Subsequently, the statewide incidence rate was estimated as the overall mean incidence rate. In addition, descriptive statistics were presented stratified by herd size (<1305, 1305–3500, and >3500), region (NCA, NSJV, and GSCA), and whether a veterinarian was involved in AMD purchase decisions. Descriptive statistics were performed using Stata 15 (Stata Corp, College Station, TX, USA).

Multiple factor analysis (MFA) was conducted separately by production type (conventional or organic). The MFA was performed to summarize the correlation structure of AMD use in adult cows and identify important principal components [19]. The first two principal components with correlation coefficients (coordinates) of 0.4 or greater were retained for interpretation [16]. The percentage of variance contributed by each group to the principal components and the correlation coefficients for the component variables within each group were estimated. For each production type, hierarchical clustering was performed on the MFA principal coordinates using the Ward’s criterion to aggregate individual dairies into relatively homogeneous subgroups (clusters) [19,20]. The identified clusters were described based on the variables that contributed the greatest to the data variability. Both MFA and hierarchical clustering were performed in R software using the “FactoMineR” package [21].

### 2.4. Comparing Antimicrobial Stewardship on California Dairies 2019 versus 2018

To describe the changes in AMD use and stewardship practices on dairies that responded to the current survey and an earlier survey in 2018 [13], responses to both surveys were compared using a two-sample test of proportions for categorical variables and two sample t-test for continuous variables in Stata.

## 3. Results

### 3.1. Herd Demographic

Survey responses were all received by mail, with no requests for a phone survey. A total of 140 (11%) survey responses from 19 of the 31 dairy producing counties in CA were received following two mailings (Figure 2). However, we received only 131 unique surveys (10.2%). The remaining nine surveys received were blank. Following the distribution of dairy herds statewide (Table 1), the greatest proportion of responses came from NSJV (41.2%) and GSCA (40.0%), followed by NCA (18.3%). Northern California dairy herds were composed of 25% Holstein, 8.3% Jersey, 8.3% crossbred, or 58.3% multiple breed herds. NSJV dairy herds were composed of 51% Holstein, 1.9% Jersey, 7.5% crossbreed, and 39.6% mixed breeds. GSCA dairy herds were composed of 67.9% Holstein, 1.9% Jersey, 1.9% crossbred, and 28.3% mixed breeds.

Respondents’ herd sizes ranged from 70 to 6000 cows/herd, with an overall average of 1348 ± 109 milking cows/herd. The average number of milking cows per respondent farm in NCA, NSJV, and GSCA was 330 ± 69, 1374 ± 163, and 1763 ± 180, respectively. A total of 86.2% surveyed dairies were conventional dairies with a mean herd size of 1507 ± 118 cows, while the remaining 13.7% were organic dairies with an average herd size of 287 ± 39.9 cows (Table 1). Responses from organic dairies (*n* = 18) were primarily from NCA (16) with one dairy in NSJV and another dairy in GSCA. The annual rolling herd average of the survey respondents’ dairies ranged from 4672 to 20,674 kg/cow, with a median of 11,589 kg/cow. The annual rolling herd average in NCA, NSJV, and GSCA was 8739 ± 516, 12,095 ± 264, and 11,761 ± 166 kg/cow, respectively.

### 3.2. Dairy Cow Health Management and Antimicrobial Use

#### 3.2.1. Dry Cow Treatment (Dry-Off Protocols) Practices

Overall, 75.2% of the respondents used blanket dry cow treatment (BDCT) with intramammary (IMM) AMDs alone (40.2%), or IMM AMDs and teat sealant (35%) at dry-off (Table 2). Of the survey respondents, 10.1% of dairies reported use of SDCT with AMDs at dry-off, specifically, IMM AMDs only (5.9%), or both IMM AMDs and teat sealant (4.2%). Approximately 12.5 ± 6.8%, 11.1 ± 4.3%, and 5.6 ± 3.2% of the respondent dairies used SDCT with IMM AMDs and/or teat sealant in NCA, NSJV, and GSCA, respectively. Survey responses showed that most dairies (65%) used the cephalosporin class of AMDs for treatment at dry-off including ceftiofur hydrochloride and cephapirin benzathine, while 41% used AMDs containing penicillins such as penicillin G procaine and cloxacillin benzathine. Only 12% of respondents indicated use of penicillin and aminoglycoside combination (Table 2).

#### 3.2.2. Dairy Cow Health Management and Vaccination Practices

The survey showed that 63.2% of respondents provided a separate fresh pen, other than the hospital pen, for their herd’s recently calved cows. In addition, 95.2% harvested colostrum from recently calved cows to feed newborn calves. Of the survey respondents, 75.8% of dairies reported vaccinating lactating cows to prevent mastitis due to coliforms, and 12.2% reported vaccinating cows to prevent *Staphylococcus* mastitis (Table 3). In addition, 37.1% of the responding dairies reported vaccinating lactating cows to prevent diarrhea in calves, 85.3% to prevent respiratory disease, and 76.7% to prevent abortion and infertility. Furthermore, 49.1%, 38.7%, and 2.6% of respondents reported vaccinating their lactating cows to protect against clostridium, pinkeye, and footrot, respectively (Table 3).

#### 3.2.3. Dairy Cow Health Protocols and Antibiotic Treatment Practices

Most respondents (93.4%) indicated they relied on a veterinarian, or a veterinarian in addition to other sources, for information about AMDs used to treat their cows (Table 4). Our results showed that 40.6% of the surveyed dairies included a veterinarian in their decision on which oral AMDs to purchase, while 45.5% of the surveyed dairies included a veterinarian in their decision on which injectable AMDs to purchase. Additionally, 48.7% of dairies included a veterinarian in their decision on which AMDs are used to treat sick cows; otherwise, decisions were made by dairy owners (65%), herd manager (57.7%), treatment crew (21.1%), milker (1.6%), and nutritionist (0.8%) (Table 4).

Approximately a quarter (24%) of the respondents indicated not having written/computerized animal health protocols (Table 4). In addition, of the respondents who reported having health protocols, 14.4% of the dairies did not rely on a veterinarian, and rather relied on dairy personnel only in developing these protocols. However, half of those that did not rely on a veterinarian for developing health protocols indicated that a veterinarian reviewed/revised the protocol, while the remaining half indicated that the protocols did not include prescription AMDs. Furthermore, 88.1%, 77.4%, and 25.8% of respondents indicated that their dairies had protocols for disease-specific treatments, vaccination schedules, and hoof trimming schedule, respectively. Most of the respondents (93.6%) reported that their disease-specific treatment protocols included milk, meat, or milk and meat withdrawal intervals. Few dairies (5.6%) did not include milk or meat withdrawal intervals in their treatment protocols, while less than 1% indicated that they were not sure about their herd health protocol details (Table 4). A total of 90.2% of the respondents reported training their treatment staff or milkers on treatment protocols for sick cows (Table 4). More than half (57.8%) of the respondents reported that a veterinarian was involved in training the treatment staff or milkers on protocols for sick cows, and that protocols were reviewed or revised once or twice a year (65.8%), every few years (10.6%), or when a new product was added (23.5%) (Table 4). Approximately three-quarters (73%) of the respondents reported including a veterinarian when reviewing their animal health protocols; the remaining dairies involved only the dairy owner and/or herd manager in health protocol development.

#### 3.2.4. Antimicrobial Drug Selection, Dosing, and Tracking Practices

Figure 3 depicts the responses of dairy producers on the importance of AMD use/indications on dairy farms. Of 125 survey respondents who completed this section, 106 (84.8%) reported that AMDs are very important to treat sick animals, while less than 4% reported that AMDs are not important to treat sick animals on their farms. The 4% that reported that AMDs were not important to treat sick cows were all organic dairies. Regarding the importance of AMDs for control of ongoing diseases, 80.8% of respondents reported that AMDs are important or very important to control the spread of ongoing disease. Similarly, 73.3% of respondents reported that AMDs are important or very important to prevent diseases in high-risk cows (Figure 3).

More than half of respondents (62.4%) reported not keeping a drug inventory log for their dairies (Table 5). For dairies who reported keeping a drug inventory log (37.6%), half of them recorded at least one piece of drug-related information, such as drug name, drug cost, quantity on hand, date of purchase, drug supplier/source, and drug expiration date. Furthermore, 59% of respondents relied on their herd veterinarian’s input for estimating AMD doses for cows. The remaining respondents who did not include a veterinarian in dose estimation (40.9%) relied on a combination of multiple factors for estimating AMD doses, including animal weight and use of a manufacturer’s labelled dosage; animal weight and use of a different dosage than the manufacturer’s label; use of a standard dose by category of animal; how sick the animal appears; and based on the disease. Approximately three-quarters of the respondents (75.8%) reported following the veterinarian’s prescription for determining the AMD treatment duration. Others (24.2%) reported using the manufacturer’s label, animal’s clinical signs, or previous experience with the drug to determine the treatment duration (Table 5). Selection of a second AMD to treat a sick animal, if the first was not successful, was based on several factors including bacterial culture and sensitivity testing (18.5%), veterinarian’s recommendation (56.3%), following the animal health protocols (32.7%), and/or previous results with the same drug (29.4%) (Table 5). Regarding the information recorded during AMD treatment, most respondents tracked treatment dates (97.6%). Milk and meat withdrawal intervals were tracked by 73.6% and 72.0%, respectively, while AMD dose and route were tracked by 54.4% and 33.6% of respondents, respectively. In addition, 7.2% of respondents recorded other information such as duration and number of treatments, and staff who administered the AMDs (Table 5).

Respondents commonly selected more than one method to track AMD treatments administered to cows. Of the survey respondents, 64.3% confirmed using computer software to track AMD treatment administered to cows. Of dairies that used a computer software to track antibiotic treatment, 42% used DairyComp 305 (Valley Agricultural Software company, Tulare, CA, USA), 11% used DHI Plus (Amelicor, Provo, UT, USA), 2% used other software such as Dairy Quest (ProfitSource, Merrill, WI, USA) and AfiFarm (Afimilk Ltd., Afikim, Israel), and 9% did not specify a software. Approximately 76% of respondents tracked AMD treatments using paper records, 44.4% reported use of markings on the cows to track AMD treatments, and 24.4% reported use of a white/chalk board or other temporary record for tracking AMD treatments in cows. Few respondents (6.3%) tracked AMD use by memory and other sources such as leg bands or a notebook.

#### 3.2.5. Antimicrobial Drug Choices for Treatment of Common Cow Diseases

This section of the survey described the AMD used for treatment of dairy cattle in California for commonly observed disease conditions including mastitis, metritis, lameness, pneumonia, and postoperative care.

##### Mastitis Treatment

Based on the survey responses, the statewide mastitis treatment incidence rate was 2.2 mastitis cases per 100 milking cow months. Approximately half of the respondents’ dairies (51.4%) relied solely on findings of abnormal milk for their mastitis treatment decisions. The remaining producers relied on a combination of factors including abnormal milk and laboratory testing (30.8%), as well as treatment while culture is pending and then modifying treatment if needed (17.7%). Approximately three-quarters (69%) of the study dairies used IMM AMD infusion to treat dairy cattle mastitis (Appendix A). Approximately 70% of respondents reported use of IMM AMDs alone, 22.1% used both IMM AMD infusion and an oral/injectable AMD, and 5.31% used only an oral/injectable AMD for treatment of mastitis in dairy cows. Fewer than 4% of respondents reported not using AMDs for treatment of mastitis. The first choice of AMD for IMM treatment of mastitis was cephalosporins (69.6%), followed by penicillins (8.9%), lincosamides (3.5%), and tetracyclines (1.7%) (Appendix A). In our study, the first choice for oral or injectable AMD for mastitis treatment was cephalosporins (5.4%), followed by tetracyclines (4.50%), penicillins (3.6%), and sulfonamides (2.7%). Our study showed that organic dairies (*n* = 18) did not use AMDs to treat mastitis. However, four dairies reported the use of non-AMD natural compounds for treatment of clinical mastitis. The use of intramammary compounds in organic dairies included PHYTO-MAST^®^ (one dairy), udder oil (one dairy), BioFresh^®^ vitamin, and mineral boluses (one dairy). Other reported approaches for treatment of mastitis on organic dairies included hand stripping of affected quarters and calf feeding (one dairy). However, two organic dairies reported used of LA200 (oxytetracycline hydrochloride) for the treatment of mastitis and pneumonia; this practice would be allowed under organic marketing if those dairies segregated treated animals and sold them to a non-organic market.

##### Metritis Treatment

Based on the survey responses, the statewide metritis treatment incidence rate was 1.17 metritis cases per 100 milking cow months. More than half of the study dairies (56.4%) reported relying on both animal-related factors (retained placenta, vaginal discharge characteristics, and twins or difficult calving) and human-derived factors (rectal temperature and rectal palpation) as their basis for treatment decisions (Appendix A). More than half of surveyed dairies (57.5%) reported using only oral/injectable AMDs to individually treat dairy cattle for metritis and 20.7% used only intrauterine AMDs, while 17.9% used both oral/injectable and intrauterine AMDs for treatment of metritis. The first choice for oral/injectable AMD for metritis treatment was cephalosporins (43.4%), followed by penicillins (13.2%), and tetracyclines (0.9%). Both tetracyclines and cephalosporins were each separately used as the first choice of AMD for intrauterine treatment by 11.5% of respondents, followed by penicillin (2.8%) (Appendix A).

##### Lameness Treatment

Based on the survey responses, the statewide lameness treatment incidence rate was 1.62 lameness cases per 100 milking cow months. Approximately half (51.7%) of the study dairies relied on both signs of lameness and hoof trimmer exam for treatment decisions, 24.1% relied on signs of lameness only, and 24.1% relied on hoof trimmer exam only (Appendix A). Approximately half (49.5%) of the study dairies used hoof treatment only (antibiotic foot wrap, heel spray, or foot bath) to treat lameness in cows, 10.4% used oral/injectable treatment only, and 38.3% used both hoof treatment and oral/injectable treatment. The first choice of AMD for hoof treatment was tetracyclines (31.2%), followed by penicillins (2.6%) and sulfonamides (0.8%). The first choice for oral/injectable AMD for treatment of lameness was cephalosporins (25.2%), followed by sulfonamides (9.9%), penicillins (6.3%), tetracyclines (0.9%), and macrolides (0.9%) (Appendix A).

##### Pneumonia Treatment

Based on the survey responses, the statewide pneumonia treatment incidence rate was 0.2 pneumonia cases per 100 milking cow months. All the surveyed dairies indicated that they rely on respiratory clinical signs (cough, difficult breathing, nasal discharge) as the basis for treatment decision for pneumonia in cows (Appendix A). The first choice reported for oral/injectable AMDs for treatment of adult cattle pneumonia was the cephalosporin class (32%), followed by penicillins (16%) and amphenicols (11%). Other dairies reported use of sulfonamides (4%), macrolides (4%), and tetracyclines (3%) as the first choice for treatment of pneumonia in adult cows (Appendix A).

##### Postoperative Care

Based on the survey responses, the statewide postoperative treatment incidence rate was 0.06 postoperative cases per 100 milking cow months. Approximately half (50.9%) of the study dairies relied on veterinary instructions as the basis for AMD treatment for postoperative care, while a third of the dairies (30.9%) reported following either veterinarian’s instructions or routinely treated after a displaced abomasum (DA) or caesarian-section (Appendix A). Less than a fifth of the responding dairies (18.2%) mentioned that they routinely treated after DA repair or caesarian-section surgery (Appendix A). Almost half of the respondents (48.9%) reported using oral/injectable AMD as part of postoperative care. The first choice for oral/injectable AMD for postoperative care was penicillins (38%), followed by cephalosporins (6.2%).

#### 3.2.6. Veterinarian–Client–Patient Relationship and Disease Diagnosis Practices

Table 6 summarizes the VCPR, and disease diagnosis practices used by the surveyed dairies. After excluding three surveys completed by veterinarians, our results showed that the majority (91.3%) of surveyed dairies confirmed they had a VCPR and, while the remaining respondents (6.3%) indicated no to this question, they also indicated that a veterinarian was included in AMD treatment decisions. Most of the dairies with a VCPR worked with a local veterinarian/clinic (93.3%), while the remaining ones (6.7%) worked with a consultant or a technical services veterinarian. Approximately 67.2% of the dairies had a written agreement signed with their veterinarian, while 32.7% had either a verbal agreement with their veterinarian and/or had not formally discussed a VCPR, but considered they had one based on veterinary care their cows received through their veterinarian. The study dairies indicated that their veterinarian observed, monitored, or discussed the health of the cows with them monthly (50%), weekly (29%), within 4 months (4.8%), and as needed (16.1%). Our study showed that 38.5% of surveyed dairies submitted non-routine samples such as milk culture, placenta, or an entire cow for necropsy and diagnosis of infectious diseases in 2019 (Table 6). One half of the dairies (49.2%) had used other on-farm diagnostic techniques or procedures, such as culture, auscultation, or lung ultrasound, to guide treatment decisions with AMD for cows in 2019 (Table 6). Only 15.1% of the survey respondents indicated that they used automated data collection systems to screen for sick cows such as sensors in ear tags, pedometers, or neck collars. The percentage of respondents reporting use of automatic data collection systems in dairies stratified by region was 19%, 11.3%, and 17.3% in NCA, NSJV, and GSCA, respectively.

### 3.3. Dairy Producer Practices and Perspectives

Approximately four-fifths (85.7%) of the surveyed dairies participated in animal welfare audit programs (Table 7). Specifically, most of the respondent dairies (85.7%) participated in animal welfare audit programs such as the National Dairy FARM Program (71.4%), Validus Dairy Animal Welfare Review Certification (7.1%), Certified Humane^®^ Program (6.35%), or the USDA organic program (0.7%). Almost half of surveyed dairies’ personnel (45.8%) participated in a dairy quality assurance program in 2019, such as the California Dairy Quality Assurance Program. Approximately 63% of the survey respondents were familiar with FDA’s term “MIADs”, i.e., they recognized that MIADs are further classified as important, highly important, or critically important drugs, and are available for livestock only via prescription or VFD pursuant to a VCPR with a licensed veterinarian. However, 37.60% ± 4.35 of the respondents indicated that they were not familiar or not sure how the FDA term relates to their dairies. The majority (96.83% ± 1.57) of surveyed dairies were aware that, since 1 January 2018, all uses of MIADs in livestock, including injectable AMDs such as Penicillin Injectable, Liquamycin^®^ LA-200 (oxytetracycline), and Tylan^®^ Injection (tylosin), as well as boluses, such as Supra Sulfa^®^ III or Sustain III (sulfamethazine), required prescription and were no longer sold OTC in CA. The majority (83.20% ± 3.9) of respondents confirmed the use of OTC and/or prescription AMDs on their dairies prior to January 2018.

Approximately half of the study dairies (47.8%) reported making changes to AMD use, which included the following: treating fewer animals with AMD (20.8%); discontinued use of one or more AMD (11.7%); using the same AMD, but at decreased dosage and duration (14.4%); or treating more animals with AMD (0.8%). The remaining respondents (52.1%) reported no changes in the use of AMD that were previously available OTC since January 2018, and all had a VCPR. A quarter (28.5%) of the study dairies confirmed usage or increased use of alternatives to AMDs since January 2018 such as vitamins, minerals, herbal remedies, and vaccines. Approximately one-third of the responding dairies (28.4%) reported making changes in management to prevent disease outbreaks or spread since January 2018; specifically, these changes included the following: improvements in vaccination programs to prevent disease (61.1%), quarantine of purchased/returning animals from offsite locations (e.g., fairs, shows, calf ranch) (2.8%), improved biosecurity (e.g., restricted traffic on operation, better isolation of sick animals, or designated separate equipment for feed and manure handling) (30.5%), and pre-purchase testing of animals before being adding to the herd (5.5%). More than a quarter (26.2%) of surveyed dairies reported decreased AMD costs on their dairies since January 2018. Additionally, 13.1% and 60.6% of the dairies reported an increase and no change, respectively, in costs of AMD on their dairies since January 2018. Furthermore, 42.8% of the surveyed dairies reported better animal health on their dairies since January 2018. Only 4.2% reported worse animal health, while 52.9% reported no change in animal health since January 2018 (Table 7).

### 3.4. Antimicrobial Drug Use Stewardship Practices

Figure 4 shows the dairy producers’ responses regarding AMD use stewardship. Of the 118 respondents, 113 (95.8%) indicated that the administration of the appropriate AMD dose, route, and duration is very important for dairy cows, while three (2.5%) and two producers (1.6%) indicated that the administration of appropriate AMD is not important and somewhat important, respectively. Regarding good record keeping on AMD treatment, 92.5%, 6.6%, and 0.8% reported that record keeping is very important, somewhat important, and not important, respectively. Of the 120 respondents, 90 (75%) producers indicated that having a VCPR is very important, while five (4.2%) producers indicated that having a VCPR is not important. The majority (95.8%) of respondents confirmed that observing AMD withdrawal periods is very important for dairy cows. More than half of producers (57.6%) indicated that using alternatives to AMDs is very important, while 28.8% and 13.5% of respondents indicated that the use of alternatives is somewhat important and not important, respectively.

Figure 5 shows the dairy producers’ responses regarding AMR stewardship. Of the 119 respondents to the statement “current antibiotic use practices in animal agriculture will make it harder to treat future livestock infections”, 20 (16.8%) respondents strongly agreed with the statement. Thirty-three (27.7%) respondents agreed, 38 (31.9%) were neutral to the statement, 21 (17.6%) disagreed, and 7 (5.8%) strongly disagreed with this statement. Of 120 respondents to the statement “antibiotic use in livestock does not cause problems in humans”, 33 (27.5%) strongly agreed, 25 (20.8%) agreed, 33 (27.5%) were neutral to the statement, 23 (19.2%) disagreed, and 6 (5%) strongly disagreed. Approximately 61% of producers disagreed or strongly disagreed that antibiotic use in livestock leads to bacterial infections in people that are more difficult to treat (Figure 5). Furthermore, approximately half of the respondents (47.1%) disagreed or strongly disagreed with the statement “any use of antibiotics may result in infections that are more difficult to treat in the future”. Our results showed that most respondents (84%) either agreed or strongly agreed to the statement “I would be willing to treat my animals with alternatives to antibiotics if they were equally effective and comparable in price” (Figure 5).

### 3.5. Stratified Analyses

A stratified analysis by herd size showed that more dairies with an average herd size >1305 milking cows used BDCT at dry off than smaller herds (Appendix A). Approximately 53%, 77%, and 100% of respondents with herd sizes <1305, 1305–3500, and >3500, respectively, reported the use of computer software to track AMD treatments administered to cows. On average, 15.1 ± 3.5, 48.1 ±5.6, and 119 ± 30.5 mastitis cases per month were reported by dairies with herd sizes <1305, 1305–3500, and >3500, respectively.

A regional analysis showed that 35.7%, 84.3%, and 94.2% of respondents used BDCT with IMM AMDs with or without teat sealants in NCA, NSJV, and GSCA, respectively, while approximately 36%, 65%, and 76% of respondents reported the use of computer software to track AMD treatments in NCA, NSJV, and GSCA, respectively (Appendix A). Across regions, on average, 5.7 ± 3.4, 38.8 ± 7.5, and 35.2 ± 6.0 mastitis cases per month were reported by dairies located in NCA, NSJV, and GSCA, respectively.

### 3.6. Multiple Factor Analysis (MFA)

The first two principal component dimensions of the MFA explained approximately 11.31% of variability in the survey responses: 7.2% and 4.1% of variance for the first and second principal component dimensions, respectively. The MFA analysis of 113 responses from conventional dairies identified seven groups (components) and 17 questions with a correlation coefficient ≥0.4 on both first and second dimensions that accounted for 80.3% of the variation in the data (Table 8). Good general practices including vaccination program accounted for 18.4% of the total variance in the data, while disease management practices (mastitis, metritis, and pneumonia) accounted for approximately 30% of the total variance in the data. Antimicrobial drug stewardship practices, AMD usage information, and producer perceptions of AMR on dairies accounted for 32.3% of the total variability in the data (Table 8).

### 3.7. Hierarchical Clustering of Conventional and Organic Dairies

Hierarchical clustering of conventional dairies. The cluster analysis partitioned conventional dairies in CA into two clusters (Figure 6); the profiles of each cluster are described in Appendix A.

Cluster 1 was predominantly composed of dairies in NSJV and GSCA, while cluster 2 was mainly represented by dairies in NCA. Regarding dairy herd size, cluster 1 included large herd sizes with a median of 1250 cows/herd, while small herd sizes with a median of 450 cows/herd were in cluster 2 (*p* = 0.004). The mean rolling herd average milk production in cluster 1 and 2 was 11,833, and 12,250 kg/cow/year, respectively. The average bulk tank somatic cell count in cluster 1 and 2 was 168,440 and 161,111 cells/mL, respectively. Most of cluster 1 dairies relied on abnormal milk as the basis for AMD treatment of mastitis, while dairies in cluster 2 relied on both abnormal milk and laboratory test results. Dairies in cluster 1 and 2 indicated that cephalosporins were the first choice AMD for IMM treatment of mastitis. For metritis treatment, most dairies in cluster 1 relied on examination (rectal palpation, taking rectal temperature) and clinical presentation (vaginal discharge, history of a difficult calving, or retained placenta) as the basis for deciding on metritis treatment. In contrast, most dairies in clusters 2 relied only on clinical presentations for metritis treatment. More dairies in cluster 1 indicated that administration of the appropriate AMD, dose, route, and duration, as well as good record keeping, were very important AMD stewardship practices in comparison with the dairies in cluster 2 (*p* = 0.001, Appendix A). Similarly, more dairies in cluster 1 strongly agreed that AMD use in livestock leads to bacterial infections in people in comparison with the dairies in cluster 2 (*p* = 0.022). In contrast, more dairies in cluster 2 agreed to the statement “AMD use in livestock does not cause problems in humans” in comparison with cluster 1 (*p* = 0.002, Appendix A).

Hierarchical clustering of organic dairies. The survey organic dairies (*n* = 18) formed a single cluster with most located in NCA (88.8%) and only two dairies in NSJV and GSCA (11.1%) and were overall composed of multiple breed herds (55.5%). The organic dairies in NCA had a median herd size of 268.7 ± 41.1 cows/herd, while the two organic dairies in NSJV and GSCA had herd sizes of 620 and 600 cows/herd, respectively. The mean annual rolling herd average of the survey organic dairies was 8018 kg/cow, with a mean bulk tank somatic cell count of 173,529 cells/mL. Out of 18 organic dairies, one dairy reported the use of treatment of all cows at dry off with an external teat sealant, while two reported selectively using a teat sealant. Most organic dairy respondents (60%) reported the use of abnormal milk as a basis for treatment of mastitis, while the remaining ones (40%) relied on laboratory testing and finding of abnormal milk to treat mastitis. Most organic dairy respondents identified that the following are very important AMD stewardship practices: administration of appropriate AMD, dose, route, and duration (90.9%); observing withdrawal periods (75%); and good record keeping on AMD treatments (66.6%). To a statement that AMD use in animals does not cause problems in humans, 58.3% of organic dairy respondents disagreed or strongly disagreed. The majority of organic dairy respondents (84.6%) showed willingness to treat animals with AMD alternatives if they are equally effective and comparable in price.

### 3.8. Comparing Antimicrobial Stewardship on California Dairies 2019 versus 2018

Our research teams conducted the same survey earlier in 2018, after SB 27 became effective in CA. As a result, we were able to compare responses from 75 dairies that completed and returned the 2018 and 2019 surveys (Figure 7; Appendix A). Our comparison showed that the percentage of respondents who strongly agreed or agreed that AMD use in livestock leads to bacterial infections in people was significantly higher in 2019 compared with 2018 (15.4% versus 5.6%, *p* = 0.020). In 2018, 40.6% of the respondents confirmed changes in management practices, while in 2019, only 24% confirmed changes in management practices (*p* = 0.035, Figure 7). The remaining survey variable comparisons were not significantly different between 2018 and 2019 responses. However, a numerically higher proportion of respondents in 2019 compared with 2018 reported that administration of the appropriate AMD, dose, route, and duration were very important (*p* = 0.051). Similarly, a numerically higher proportion in 2019 compared with 2018 kept a drug inventory log, had a current VCPR, observed better farm animal health, perceived use of AMD alternatives as very important, or strongly agreed or agreed that any use of AMD may result in infections (Appendix A).

## 4. Discussion

The primary objective of the current survey was to explore AMD use and stewardship practices on California dairies after implementation of SB 27. California is the leading dairy producing state in the United States, with over 1.7 million dairy cows producing 18.5% of the nation’s milk supply [18]. Knowing the impact of the VFD and SB 27 regulations on management and AMD use on CA dairies can help to guide future outreach efforts and form a baseline for future comparison. The current survey achieved an 11% response rate from 61% of the dairy-producing counties in CA. Producers who participated in our survey were distributed across the three milk sheds in CA, specifically, 18% in NCA, 41% in NSJV, and 40% in GSCA. Our response rate was similar to previous dairy surveys conducted in CA [16,22]. Most survey respondents (60%) were dairy owners, with a small percentage of the surveys completed by managers (19%) or owner/herd manager (18%). Only few (3) surveys (2.3%) were answered by the herd veterinarian, similar to an earlier University of California Cooperative Extension survey [22].

### 4.1. Herd Demographics

Similar to an earlier antimicrobial stewardship survey of CA dairies, the current survey showed that the state’s organic dairies were mainly located in NCA, while most of the conventional large herd sizes were located in NSJV and GSCA [13]. Like previous reports, the current survey identified fewer organic dairies in NSJV and GSCA [16,22]. According to USDA organic regulations (7 CFR §205), for farms to be eligible for organic certification, they must follow the following standards: dairy animals should be under continuous organic management from the last third of gestation, or under continuous organic management beginning no later than 1 year prior the production of milk, for milk products to be represented as organic. Use of animal drugs, including hormones, to promote growth is prohibited in organic dairies. Our survey revealed that 53.1% of surveyed dairies in CA were composed of only Holsteins, a lower estimate compared with previous reports (65% Aly et al. [23] and 77% in Love et al. [16]).

### 4.2. Dairy Cow Health Management and Vaccination practices

Blanket dry cow treatment is the use of a long-acting IMM AMDs in the four quarters of all cows at dry off. In contrast, selective dry cow treatment (SDCT) is an approach that targets specific cows affected by clinical or subclinical mastitis, and hence may benefit from such AMD treatment. The current survey revealed that 75.2% of respondent dairies used BDCT. Approximately half (40.1%) of these dairies used IMM AMDs alone, while the remaining (35%) used both IMM AMDs and teat sealant. The USDA Dairy 2014 Study [24] showed that 93% of cows nationwide were treated with dry cow IMM AMDs. Lower estimates for BDCT were reported in Europe; Bertulat et al. [25] surveyed the commercial dairy farms in Germany and found that BDCT was carried out on 79.6% of surveyed dairies. Vilar et al. [26] surveyed Finnish dairies and found that 78% reported using SDCT, 13% of farms applied BDCT, and 9% did not use any DCT. The practice of BDCT is a widely adopted component of the dry cow mastitis control plan, which involves infusing all quarters of all cows at dry off with AMDs. Historically, BDCT has played an important role in reducing the prevalence of contagious mastitis [27]. However, owing to the emergence of AMR and more restrictive use of AMDs, SDCT may be a recommended option [28]. Currently, the European Commission has restricted the prophylactic use of AMDs and requires susceptibility testing of pathogens before any AMD treatment [29]. Further research is necessary to develop effective methods for DCT to avoid AMR, such as SDCT algorithms [30].

According to our survey, most dairies in NSJV and GSCA used BDCT more than dairies in NCA. Such findings may be attributed to the fact that most organic herds (smaller herds) are located in NCA and have restrictions concerning dry cow treatments, while the largest herds were conventional and located in NSJV and GSCA. In terms of treatment at dry off, the findings from our CA survey agree with nationwide estimates that the majority of dairy operations (58.1%) administered Cefa-Dri/Tomorrow, and approximately one-fourth of operations administered Spectramast DC or Quartermaster^®^ as IMM AMD at dry off [24]. Regarding vaccination for mastitis, 75.9% of our respondents reported vaccinating lactating cows to prevent coliform mastitis (e.g., *Escherichia*, *Klebsiella*, and *Enterobacter*). Furthermore, 88.5% of our respondents reported the administration of at least one type of vaccine to prevent disease in cows. The USDA 2014 Dairy study [24] indicated that more dairy operations in the Western United States (86.3%) vaccinated their lactating cows for any disease compared with dairies in the Eastern United States (72.6%). Vaccines and other AMD alternatives can help minimize the need for AMDs by preventing and controlling infectious diseases in animal populations [31]. Several studies have demonstrated that the use of various bacterial as well as viral vaccines in animals can result in a significant reduction in AMD consumption [32].

Our study showed that California dairy producers accessed several sources to obtain information about AMDs used to treat sick cows, including drug label, industry magazines, pharmaceutical company representatives, and the Food Animal Residue Avoidance Databank (http://www.farad.org/; accessed on 9 July 2021), among other websites and sources. Overall, 92% of the respondents relied on veterinarian and/or other sources for information regarding AMD use. These results are consistent with a USDA (2018) survey [33], which confirmed that 96.2% of U.S. dairy producers consulted a veterinarian or relied on a drug label created by a veterinarian in their treatment decisions. Previous surveys indicated that dairy producers preferred to receive AMD-related information from their veterinarians as a trusted source of information [10,12].

Our study showed that 79% of respondent dairies confirmed having written/computerized animal health protocols for cows. Specifically, 85% of respondents with written/computerized health protocols reported the protocols were developed by veterinarians, while 14% reported were developed by dairy personnel. The study’s percent of dairies who confirmed written/computerized animal health protocols was greater than the 60.9% reported by the USDA survey in 2018 [33]. A survey of South Carolina dairies [12] estimated that 32% of farmers had written protocols for diagnosing and treating common medical conditions and attributed such a low percent to the lack of farmers’ time and limited finances. Our survey results showed that most of the respondents in California had written protocols, as compared with South Carolina and Pennsylvania’s survey responses [34]. Recently, the American Veterinary Medical Association has endorsed guidelines for rational and prudent uses of AMDs in cattle, including the encouraging of veterinarians to provide written or computerized treatment protocols to their clients that describe indications, meat and milk withdrawal times, and instructions for AMD use in the production facility.

The survey results also showed that more than half of respondents’ dairies (62.40%) did not keep a drug inventory log. Similarly, a 2018 survey of Tennessee dairy producers [35] showed that 42.5% of the responders did not keep written records of medicated feeds purchased in the framework of a VFD and that 7.5% were not sure about the question. Important benefits of a drug inventory to producers include quantification of AMD usage in dairy herds [36]. Proper records that fully document an animal’s treatment history and a farm’s AMD use can help avoid drug residue violations and related penalties.

Regarding producer responses on AMD use for dairy cows, 60.8% indicated that AMDs are important or very important to prevent diseases in high-risk cows, which agrees with a New York survey showing that most conventional farmers believed that their cattle’s health would suffer if AMD use were further decreased [7]. The World Health Organization (2017) published guidelines restricting the routine use of MIADs in food animals. Specifically, the WHO recommends restricting MIAD use for growth promotion and disease prevention in food animals [37]. While use of AMD for growth promotion is prohibited in the United States, their use for disease prevention is still permissible. More research and extension education are needed to guide dairy producers on disease prevention, specifically exploring the use of a risk assessment approach in control of AMR, alternatives to antimicrobial drug use, and specific guidelines for sustainable use of MIADs in food animals for prophylactic and metaphylactic purposes.

Approximately 65% of the respondents confirmed using computers to track drug treatments, specifically through dairy farm software. The current estimates contrast with an earlier CA survey’s estimate that 35% of respondents used computers to record drug treatments [38]. Although such a difference could be due to different dairies responding within the entire population of California dairies, the difference could also be due to an actual increase in software to track AMD treatments in California. A USDA study [39] reported that 98% of large herds (500 or more head) used a computer record-keeping system, and that more than 20% of these dairies reported using a computer record-keeping system reported use of DairyComp 305 (29.8%), PCDART (21.5%), or DHI Plus (24.9%). Similarly, our results showed that the majority (54%) of the dairies that reported using computer records were large herds (500 or more cows), while the remaining 10% of dairies using computer records were small herds (less than 500 cows).

Approximately 40% of our survey respondents submitted non-routine samples for the diagnosis of infectious diseases in 2019. Outreach curricula should include information for producers on the importance of utilizing disease diagnostics for herd health management. In contrast to the current CA survey, a Tennessee survey [31] found that 11.6% of dairy cattle producers used bacterial culture and sensitivity testing most of the time to determine the cause of disease and select the appropriate AMD. Precision farming is the automation and application of sensor systems and information technology in livestock systems to recognize and measure behavioral outcomes, disease, and fertility in individual and animal cohorts [40]. Only 15.1% of our survey respondents reported using automated data collection systems to screen cows for illness. An increase in the use of automated health monitoring systems is a great opportunity for veterinarians to expand their role in antimicrobial stewardship on dairies. Data collection technologies have only been adopted on a small proportion of farms globally [41,42,43]. However, precision farming is increasingly providing producers with the means to reduce labor requirements and facilitate management of large herds [44]. A 2015 Australian survey found that dairy farms with more than 500 cows had a significantly higher adoption of precision technologies compared with farms with fewer than 500 cows [45].

### 4.3. Antimicrobial Drug Choices for Treatment of Common Cow Diseases

Based on the Dairy 2014 survey [24], clinical mastitis was detected in about one-quarter of all cows (24.8%) during 2013. In most studies, the median incidence of clinical mastitis was around 20–25 cases per 100 cows per year, or the equivalent of 1.9 cases per 100 cow months [46,47]. Our survey findings agree with the aforementioned estimates; specifically, we estimated 2.2 mastitis cases per 100 milking cow months, or the equivalent of a 26.4% cumulative incidence proportion per year. In the present study, 51.4% of the respondents reported that they rely only on the finding of abnormal milk to treat cows for mastitis, while the remaining respondents (49%) relied on abnormal milk and/or laboratory testing. A USDA study [24] reported that, on 40.6% of operations nationally, mastitis treatments were guided by culture and antimicrobial sensitivity testing. Although basing mastitis treatment on abnormal milk is a common practice on dairies, these findings highlight the importance of outreach to guide producers on stewardship practices relevant to mastitis control, including the use of rapid diagnostic tests and their cost–benefit prior to initiating treatment with AMDs. Our study identified that about 70% of the respondents used IMM AMD infusion to treat mastitis. Such findings agree with a USDA report [24] that 89% of respondents used IMM AMDs for treatment of mastitis. Furthermore, our survey showed that mastitis is the most common cause of AMD treatment in dairy cows, a finding that agrees with a United Kingdom survey [48].

A USDA study [24] reported that 73% of dairies nationwide used cephalosporins as the primary AMD for treating mastitis, followed by penicillins, lincosamides, and tetracyclines. Similarly, cephalosporins were the first choice of IMM AMD for treating mastitis by our study respondents in California. In contrast to our study, ampicillin and oxytetracycline were the most common AMDs administered to cows with clinical mastitis in Wisconsin [49]. In our study, three dairies reported the use of sulfadimethoxine as their first-choice AMD for mastitis, and one dairy reported use of sulfadimethoxine as their second choice. Sulfadimethoxine is labeled only for treatment of pneumonia or foot infections and no extra-label usage of this compound is permitted for treatment of bovine mastitis [50]. Other studies similarly reported the use of sulfadimethoxine for the treatment of mastitis on eight dairy farms [34,51]. Assuming such entries were not erroneous owing to misunderstanding the question or human error, further outreach about following the AMD label is needed. In addition, two dairies reported use of oxytetracycline for IMM treatment of mastitis; given that this drug is not available as IMM infusion, the survey respondents may have mistakenly selected this route instead of injectable.

Organic dairies in our survey did not report using AMDs to treat mastitis, in agreement with findings from another study [51]. However, three organic dairies in our survey reported the use of non-antimicrobial natural compounds for the treatment of clinical mastitis. Pinedo et al. [52] tested the efficacy of PHYTO-MAST^®^ for IMM treatment of clinical mastitis, but found no significant effect on clinical mastitis resolution at day 4 post-treatment; however, a reduction in time to clinical recovery was reported. According to USDA organic regulations (7 CFR §205), the use of AMDs is only allowed in the case of emergency to save the life of the animal or to prevent suffering. If organic producers used AMDs, they must record the event in their health records, notify their certifier, segregate the animal to prevent contamination of organic products, and sell the animal to a non-organic market.

Our survey results showed that 21% of responses reported the use of only intrauterine AMDs for the treatment of metritis. Currently, in the United States, there are no approved AMDs for intrauterine administration in dairy cows. However, ceftiofur hydrochloride, a broad-spectrum third-generation cephalosporin, is approved for parenteral administration for treatment of metritis in dairy cows. Assuming the respondents’ report of intrauterine administration of ceftiofur is not in error, further outreach and extension are needed to educate producers on this unapproved route of administration.

In our study, more than half of the surveyed dairies used only oral/injectable AMDs to treat cows for metritis, with their first choice being cephalosporins, followed by penicillins and tetracyclines. Similarly, Dirillich et al. [53] reported that AMDs commonly used for the treatment of puerperal metritis included penicillin, cephalosporins, or a combination of ampicillin with oxytetracycline or cloxacillin.

The average number of cows treated for lameness in our study was 28.4 ± 6.5 cows/month. The USDA Dairy 2014 study [33] reported that 16.8% of cows were affected by lameness. The first choice for oral/injectable AMD for treatment of lameness in our study was cephalosporins (25.2%), followed by sulfonamides (9.9%), penicillins (6.3%), tetracyclines, and macrolides (1.8%). The Dairy 2014 study also reported that third-generation cephalosporins were used as the primary AMD for more than half of the cows treated for lameness, respiratory disease, and diarrhea. In contrast, tetracyclines were used as the primary AMD for lameness on 11.4% of dairy farms [33].

The average number of cows treated for pneumonia in our study was 3.0 ± 0.83 cows/month. A USDA Dairy study [33] reported that 2.8% of cows were affected by pneumonia. Our survey results show that 100% of our respondents relied on observable clinical signs to initiate the AMD treatment of pneumonia. Prior reports [51,54] showed that estimates for the quantity of AMDs used on dairy farms have been based on the individual producers’ perceptions of disease. The utilization of a novel assessment tool, such as the California BRD scoring system [55] designed for calves, can help producers monitor herd prevalence before and after interventions as well as the judicious use of AMD; a similar scoring system may be needed for adult cows. Our results highlight the need for education and training of dairy producers and employees on diagnostic criteria for initiation of AMD therapy. Approaches such as a risk assessment tool for bovine respiratory disease should be explored for antimicrobial drug use [56]. In agreement with our results, the USDA Dairy 2014 study [33] reported that cephalosporins were the primary AMD used for respiratory disease in adult cows.

### 4.4. Dairy Farmer Practices and Perspectives

Approximately half of responding dairies’ personnel (45.8%) participated in a dairy quality assurance program (DQAP) in 2019, such as California DQAP, FARM, and Validus. Quality assurance programs are designed to improve dairy cattle production and welfare through assessments and routine monitoring. A USDA survey [39] found that approximately half of the surveyed operations (45.9%) participated in a quality assurance program sponsored by the state, a milk cooperative, or a national association or entity. Like our findings, a different California survey found that 60% of the dairy producers would consider joining DQAP, whereas 9% indicated that they would not [38].

Our study indicated that 37.6% of the respondents were not familiar with or not sure how the FDA “MIADs” term relates to their dairies. These results highlight the urgent need for education and training of producers and dairy farm employees on prudent use of AMDs that are considered MIADs. Similarly, a Tennessee survey of AMD practices showed that 22.7% of dairy cattle producers reported that they were not concerned about AMR, and that 6.8% did not rate their degree of concern about AMR because they were not familiar with the concept of AMR [35].

Approximately 48% of our respondents reported making changes to AMD use in response to SB 27, which could indicate a successful impact of SB-27 legislation on the judicious use of AMDs on CA dairies. In the Netherlands, AMD usage in dairy cattle decreased by 47% in the period 2009–2015 after intense cooperation between the dairy industry stakeholders (representatives of the producers’ organizations, the dairy and meat plants, and the veterinarians, as well as technical experts), and the introduction and implementation of the farm health plan and farm-specific treatment protocols, resulting in changes in AMD use [57].

Approximately 45% of our respondents agreed or strongly agreed that current AMD use practices will make it harder to treat future livestock infections. A negative perception of AMD use has been reported previously; a survey of dairy veterinarians in Ontario, Canada found that the majority of respondents (81%) agreed with the question “Do you feel AMD use at the current levels within the dairy industry, is a contributor to decreased AMD efficacy” [58]. Furthermore, 61% of our respondents disagreed or strongly disagreed that AMD use in livestock leads to bacterial infections in people that are more difficult to treat. The majority of the respondents (86%) of the Canadian survey reported some level of disagreement or no opinion to the question asking whether AMD use in dairy cattle could contribute to AMR in humans [58]. A survey of dairy producers in Michigan and Ohio reported that 29% agreed that AMD use in agriculture makes it harder to treat infections in livestock in the future and only 7% agreed that AMD use in livestock leads to bacterial infections in people that are difficult to treat [11]. A survey of Washington dairy producers reported that 74% of producers agreed that “AMDs become less effective the more they are used” and more than half of the respondents (59%) agreed that AMD use in food animals could affect human health [59].

Of our survey participants, 21% agreed and 28% strongly agreed with the statement “antibiotic use in livestock does not cause problems in humans”. Similarly, McDougall et al. [60] also found that, while producers understood there was a risk of AMR occurring on dairy farms, they did not agree that their use of AMDs was associated with risks of AMR within human populations, or on other farms. Good AMD stewardship is paramount to animal and public health and the continuous effectiveness of these valuable compounds. Further outreach to producers is required to increase understanding of the AMR issue and the important role they play in protecting public health.

In the present study, most respondents (84%) showed a willingness to treat their animals with alternatives to AMDs if they were equally effective and of comparable cost. The willingness of dairy producers to use AMD alternatives highlights the need for more research on AMD alternatives that can be used by the dairy industry. Habing et al. [11] reported that alternative therapies such as probiotics, garlic, aloe, and “other herbal therapies” were used by both organic and conventional dairy producers in Michigan and Ohio for the treatment of calf diarrhea. However, more research is needed to ensure the effectiveness and safety of such products for dairy cattle.

#### Clustering of CA Dairies by Antimicrobial Stewardship Practices

Our factor analysis identified important questions or characteristics that, when grouped, can make up the components of a risk assessment tool for AMR on dairies, an approach that has been used successfully in the past to develop risk assessments for paratuberculosis in cattle [61] and, more recently, for respiratory disease in preweaned dairy calves [23]. Such a risk assessment approach would focus on management practices known to increase the risk of AMR development; each practice is scored proportional to their overall contribution to antimicrobial resistance at the herd level. A data-driven approach based on longitudinal follow-up studies would be necessary to estimate the magnitude of the association between management practices and the risk of AMR measured phenotypically and genotypically.

Cluster analysis has also been used in previous California surveys, including one to develop the risk assessment tool for bovine respiratory disease in calves [16,62]. In the current study, cluster analysis identified that cow management practices on NCA dairies were different than practices on dairies in other regions of CA. Dairies in the NCA region were significantly smaller in herd size, had a smaller proportion of Holsteins, and were mostly organic when compared with dairies in the remaining two regions. Our findings agree with an earlier CA survey conducted in 2018 [13] that classified conventional CA dairies into two clusters with large-scale conventional dairies located mainly in GSCA and NSJV, and mid-sized conventional dairies in NSJV and NCA. Stratified analyses by region showed that approximately 84% of respondents in NSJV have written/computerized health protocols, while approximately 48% and 41% of respondents in NCA and GSCA, respectively, have written/computerized health protocols. More outreach efforts on the importance of having written/computerized health protocols are necessary to improve antimicrobial stewardship.

### 4.5. Antimicrobial Stewardship Practices Immediately after SB 27 (2018) versus a Year Later (2019)

There was a significant difference in management and AMD stewardship practices in CA dairies comparing survey data obtained in 2019 with their responses from 2018. However, the survey respondent’s agreement with the statement that the use of AMDs in livestock leads to bacterial infections in people was significantly higher in 2019 compared with 2018 shows increased awareness of AMR and its importance. These results may highlight the increase in stewardship efforts after SB 27 and the benefits of research and education on judicious use of AMD and implementation of stewardship practices. Comparing our results with the same survey conducted in 2018 [13], more respondents (41%) reported changes in management practices during 2018, immediately after implementation of SB 27, than in 2019, when only 24% of respondents reported changes in management practices. The difference may reflect that no further changes were adopted once the same respondents initially modified their management practices regarding MIADs in response to new legislation. In addition, a proportional increase in the number of dairies that reduced their AMD use in 2019 (49%) versus 2018 (44%) may indicate that implementation of SB 27 continues to have an impact on AMD use, albeit at a constant rate. Our findings agree with an earlier survey conducted in 2018 [13] that found that disease management practices (mastitis, metritis, and pneumonia), herd demography, AMD usage information, AMD use stewardship practices, and producer perceptions of AMR on dairies were important components that accounted for significant variance among dairies. Therefore, extension and outreach efforts should focus on those components to improve antimicrobial stewardship. 

Overall, AMD stewardship programs within dairies may choose to focus on reductions in disease incidence, utilization of AMD alternatives, or more stringent diagnostic criteria for initiating AMD. Further monitoring of AMD use and stewardship practices, as well as associations between AMD use and AMR in California, will help to evaluate the effects of the implementation of SB 27.

### 4.6. Limitations

Response rates to surveys, including those administered by mail, have declined in recent years, which is a limitation of the present study method [63,64]. While the response rate to this survey was relatively low overall, the result is consistent with other mailed surveys conducted in CA [16,22] and another survey of Canadian dairy farms [65]. Furthermore, response rates stratified by region in our survey were like the regional distribution of herds as reported in a California Department of Food and Agriculture report [18]. Therefore, potential bias due to regional difference was minimal, as all three regions were proportionately represented in our survey. Questionnaires are retrospective by nature, hence our survey is subject to recall bias. Another limitation associated with surveys in general is the possibility for misinterpretation of questions, and likewise for responses. To avoid this, we first tested the questionnaire with colleagues and stakeholders; in addition, we used as many closed-ended questions as possible with ‘other (please specify)’ options, as needed. In addition, responses to survey questions may be influenced by the respondent’s role, perception, and attitude towards a particular question or topic; specifically, three surveys returned were completed by veterinarians. A few respondents chose not to answer all the survey questions, resulting in a variation in the frequency of responses by question. Incomplete surveys may have been the result of the comprehensive nature of the survey, with respondents not knowing all the answers. Finally, as with similar surveys, responses could not be verified and, in the case of our survey, lacked an assessment of AMR on the surveyed dairies.

## 5. Conclusions

Our survey successfully characterized a representative convenience sample of CA dairies’ AMD use, disease management, and AMD stewardship practices. The convenience sample mirrored closely the regional distribution of California dairies in terms of herd size, breed distribution, milk production, and organic versus conventional status. The results of our study describe a detailed account of the demographic parameters, dry-off protocols, disease management, and AMD stewardship practices of many CA dairies in the second year immediately after implementation of SB 27. The results serve as a roadmap for future extension and outreach efforts to advance AMD stewardship on California dairies and update the baseline data provided by an earlier survey on the same subject [13]. Future extension and outreach on antimicrobial stewardship is needed to guide the dairy industry and, specifically, to expand the use of diagnostic tests to confirm the need for AMD use, more widespread application of written/computerized health protocols, and training of farm personnel on the problem of AMR and associated outcomes on animals and humans. Finally, further research is needed to develop and optimize AMD alternatives given that respondents showed interest in adopting such alternatives if they are as effective as AMDs.

## Figures and Tables

**Figure 1 microorganisms-09-01507-f001:**
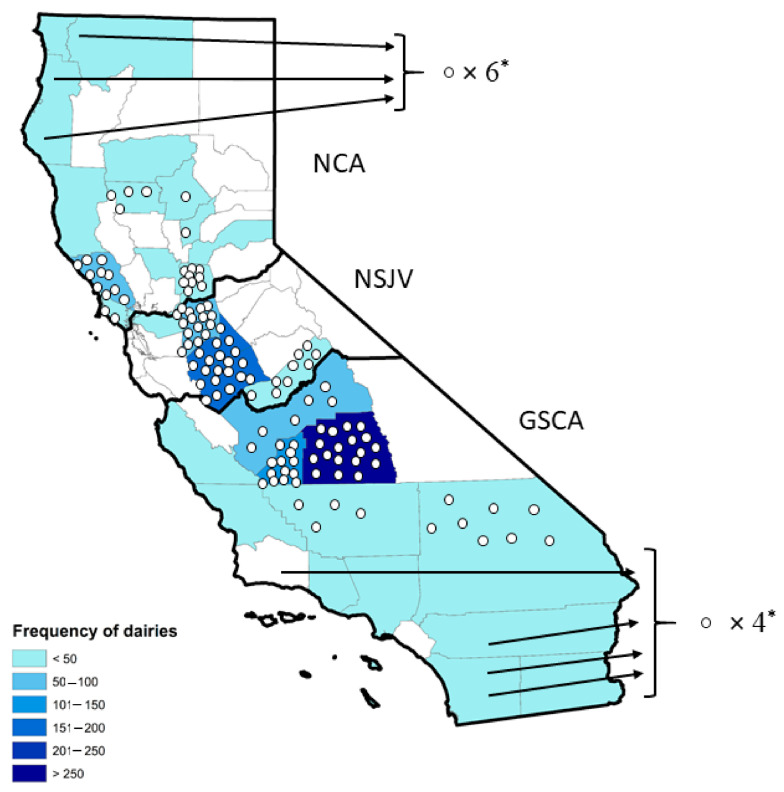
Map of counties in Northern California (NCA), Northern San Joaquin Valley (NSJV), and Greater Southern California (GSCA) regions used for comparison of survey responses. White dots represent the received surveys from each county. * The locations of respondents by county are censored to maintain confidentiality. Survey responses identified in the center of the county and do not reflect actual respondent location.

**Figure 2 microorganisms-09-01507-f002:**
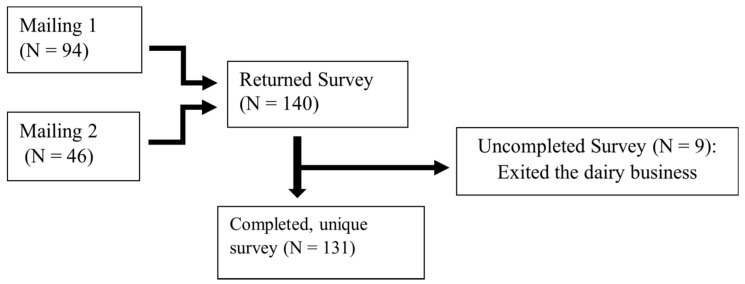
Summary of returned and completed surveys on antimicrobial drug use in adult cows mailed in 2019 to 1282 licensed grade A California’ dairies.

**Figure 3 microorganisms-09-01507-f003:**
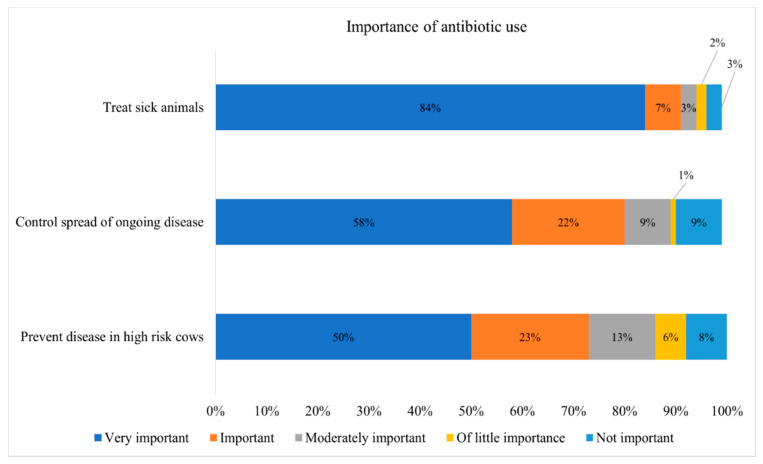
California dairy producers’ responses on the importance of antibiotic use for treatment, control, and prevention of diseases in dairy cows.

**Figure 4 microorganisms-09-01507-f004:**
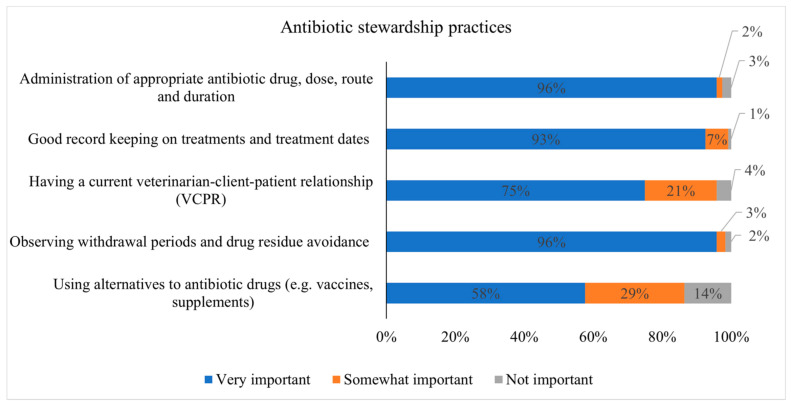
California dairy producers’ responses regarding antimicrobial drug use stewardship practices.

**Figure 5 microorganisms-09-01507-f005:**
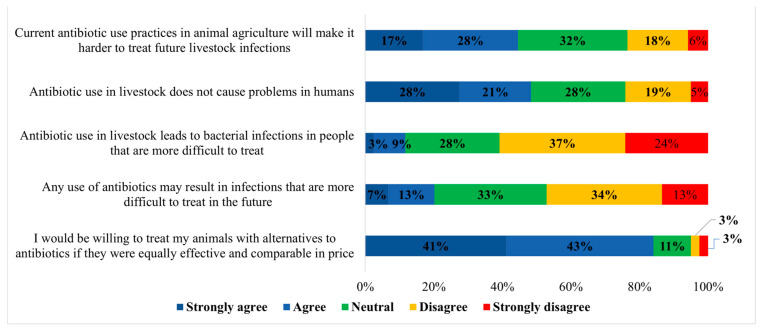
Findings of agreement level on statements related to antimicrobial resistance in dairy cows in California.

**Figure 6 microorganisms-09-01507-f006:**
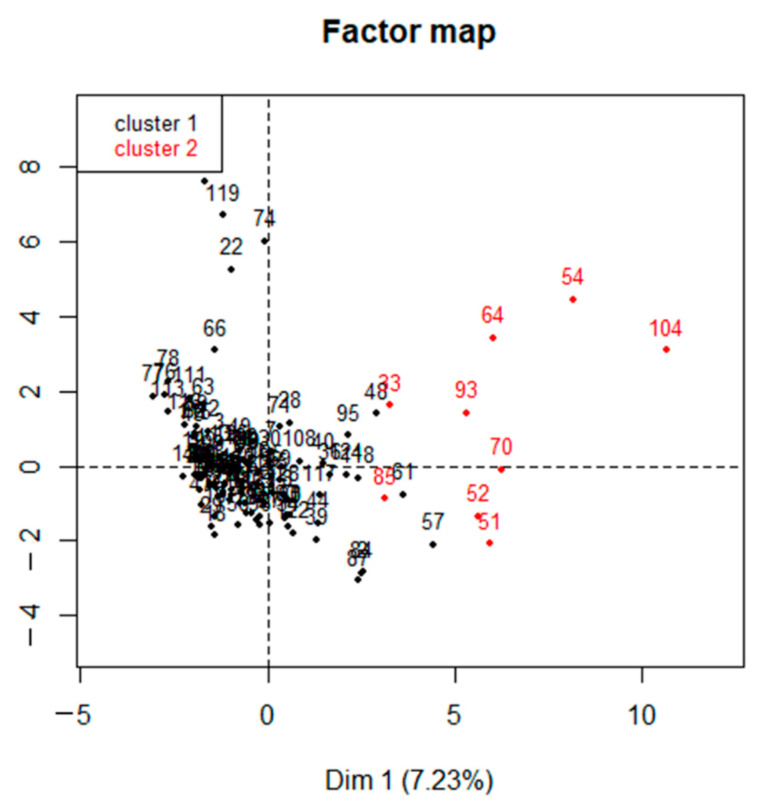
Representation of the two clusters identified using the results of the multiple factor and hierarchical clustering analyses of the 113 survey responses from conventional dairies in CA during 2019.

**Figure 7 microorganisms-09-01507-f007:**
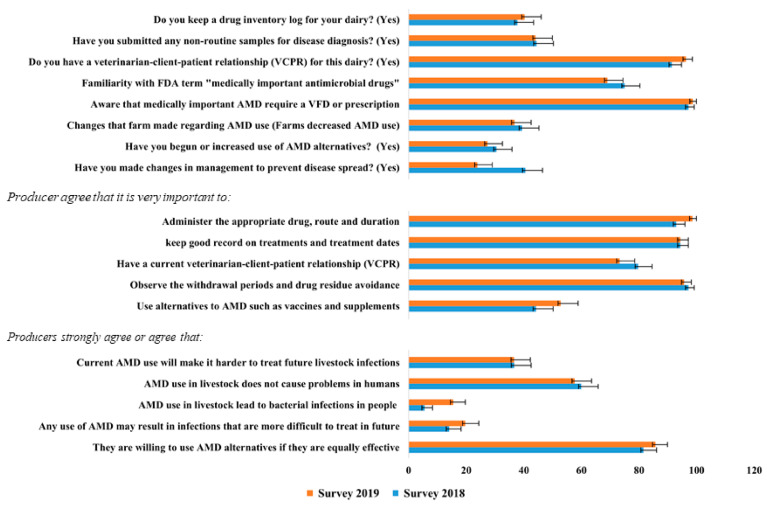
Responses of California dairy producers (*n* = 75) to two surveys conducted in 2018 and 2019 on antimicrobial drug (AMD) use and stewardship practices in adult cows.

**Table 1 microorganisms-09-01507-t001:** Summary of herd information from responses to a mailed questionnaire on antimicrobial drug (AMD) use in adult cows on California dairies during 2019.

				95% Confidence Limits	
Herd Demographic	Missing Data	*n*	Estimate (%)	Lower	Upper
**Respondent role**	0	131			
Manager		25	19.1	13.2	26.8
Owner		79	60.3	51.6	68.4
Owner-Manager		24	18.3	12.5	26.0
Veterinarian		3	2.3	0.7	7.0
**Region**	0	131			
Northern California (NCA)		24	18.3	12.5	26.0
Northern Central (NSJV)		54	41.2	33.1	49.5
Greater southern California (GSCA)		53	40.5	32.3	49.2
**Management**	0	131			
Certified organic		18	13.7	8.8	20.8
Conventional		113	86.3	79.2	91.2
**Herd size (milking cows/herd)**	1	130			
<1305		78	60.0	51.3	68.2
1305–3500		44	33.9	26.2	42.5
>3500		8	6.2	3.1	11.9
**Rolling herd average (kg/cow)**	11	120			
<10,880		32	26.7	19.4	35.4
≥10,880		88	73.3	64.6	80.6
**Bulk tank somatic cell count (cells/mL)**	2	129			
<100,000		15	11.6	7.1	18.5
100,000–199,999		80	62.0	53.2	70.1
≥200,000		34	26.4	19.4	34.7
**Breed**	1	130			
Holstein (100%)		69	53.1	44.4	61.6
Jersey (100%)		4	3.1	1.14	8.0
Crossbred (100%)		7	5.4	2.6	11.0
Mixed breed		50	38.5	30.4	47.2

**Table 2 microorganisms-09-01507-t002:** Summary of dry cow treatment practices from responses to a questionnaire on antimicrobial drug (AMD) use in adult cows on California dairies during 2019.

				95% Confidence Limits	
Dry Cow Treatment Practice	Missing Data	*n*	Estimate (%)	Lower	Upper
**Blanket treatment of all dry cows** ^1^	14	117			
Yes		97	82.9	74.8	88.8
No		20	17.1	11.2	25.2
**Blanket treat all dry cows** **with:**	14	117			
Intramammary AMD		47	40.2	31.3	49.1
Intramammary AMD + Teat sealant		41	35.0	26.4	43.7
Teat sealant only ^2^		9	7.7	2.8	12.5
**Selective dry cow treatment**	14	117			
Yes		19	12.2	10.5	24.2
No		98	83.2	75.8	89.5
**Selective dry cow treatment** **with:**	14	117			
Intramammary AMD		7	5.9	1.6	10.2
Intramammary AMD + Teat sealant		5	4.2	0.6	7.9
Teat sealant only ^2^		7	5.9	1.6	10.2
**AMD used in dry cow treatment (blanket or selective) ^3^**	31	100			
Cephalosporins		59	59.0	49.0	68.3
Penicillins		23	23.0	15.7	32.4
Cephalosporins or Penicillins		2	2.0	0.5	7.8
Cephalosporins or Penicillins or Aminoglycosides		4	4.0	1.5	10.4
Penicillins and Aminoglycosides		12	12.0	6.9	20.1

^1^ Blanket dry cow treatment = an approach to treat all quarters of every cow at drying-off either with IMM antibiotics infusion with or without a teat sealant. ^2^ Teat sealant is not an AMD. ^3^ The AMD classes reported by the survey respondents included the following: Cephalosporins (ceftiofur hydrochloride (Spectramast^®^), cephapirin benzathine (Tomorrow^®^)); Penicillins (cloxacillin benzathine (Orbenin^®^, Boviclox^®^), penicillin G procaine/novobiocin (Albadry^®^)); Penicillins–Aminoglycosides combinations (penicillin G procaine/dihydrostreptomycin (Quartermaster^®^)).

**Table 3 microorganisms-09-01507-t003:** Summary of dairy cow vaccination practices from responses to questionnaire on antimicrobial drug use in adult cows on California dairies during 2019.

				95% Confidence Limits	
Vaccination Practice	Missing Data	*n*	Estimate (%)	Lower	Upper
**Mastitis vaccine [*Coliform*]**	15	116			
Yes		88	75.9	67.1	82.9
No		28	24.1	17.1	32.9
**Mastitis Vaccine [*Staphylococcus*]**	16	115			
Yes		14	12.2	7.3	19.6
No		101	87.8	80.4	92.7
**Diarrhea/Scours vaccine [*E. coli*, Rota, Corona]**	15	116			
Yes		43	37.1	28.7	46.3
No		73	62.9	53.7	71.3
**Respiratory disease vaccine**	15	116			
Yes		99	85.3	77.6	90.8
No		17	14.7	9.2	22.5
**Abortion/infertility vaccine [Leptospirosis, BVD** ^1^ **]**	15	116			
Yes		89	76.7	68.0	83.6
No		27	23.3	16.4	32.0
**Pinkeye vaccine**	15	116			
Yes		45	38.8	30.3	48.1
No		71	61.2	51.9	69.7
***Clostridium* vaccine**	15	116			
Yes		57	49.1	40.0	58.3
No		59	50.9	41.7	60.0
**Footrot vaccine**	15	116			
Yes		3	2.6	0.8	7.8
No		113	97.4	92.2	99.2
***Salmonella* vaccine [SRP** ^2^ **]**	15	116			
Yes		4	3.5	1.3	9.0
No		112	96.6	91.1	98.7

^1^ BVD = bovine viral disease; ^2^ SRP = siderophore receptors and porins (SRPs).

**Table 4 microorganisms-09-01507-t004:** Summary of responses to questions about dairy cow health protocols and treatment practices from responses to a questionnaire on antimicrobial drug (AMD) use in adult cows on California dairies during 2019.

				95% Confidence Limits	
Animal Health Protocol and Antimicrobial Drug Use Practices	Missing Data	*n*	Estimate (%)	Lower	Upper
**Sources of information on AMD used to treat cows**	8	123			
Veterinarian only		113	91.9	85.4	95.6
Veterinarian + others ^1^		2	1.6	0.4	6.4
Others only		8	6.5	3.3	12.6
**Who decides which oral AMD to purchase?**	8	123			
Include veterinarian		50	40.7	32.2	49.7
Dairy personnel only		73	59.4	50.3	67.8
**Who decides which injectable AMD to purchase?**	19	112			
Include veterinarian		51	45.5	36.4	55.0
Dairy personnel only		61	54.5	45.1	63.6
**Who decides which AMD to treat sick cows?**	8	123			
Include veterinarian		60	48.8	40.0	57.7
Dairy personnel only		63	51.2	42.3	60.1
**Use of written/computerized health protocols**	13	118			
Yes		90	79.3	67.6	83.2
No		28	23.7	16.6	32.4
**Who developed the protocols?**	41	90			
Include veterinarian		77	85.6	76.5	91.5
Dairy personnel only		13	14.4	8.5	23.5
**Health aspects for which protocols are used**	43	88			
Therapeutic ^2^ + prophylaxis ^3^		74	84.1	74.7	90.4
Therapeutic only		14	15.9	9.6	25.3
**Therapeutic protocols include the following information**	47	84			
Milk and meat withdrawal interval		74	88.1	79.0	93.6
Milk or meat withdrawal interval		5	5.6	2.5	13.7
No milk or meat withdrawal interval		5	5.6	2.5	13.7
**Who has access to the protocols?**	42	89			
Include veterinarian		66	74.2	63.9	82.3
Dairy personnel only		23	25.8	17.7	36.1
**Are treatment staff trained on protocols for sick cows?**	39	92			
Yes		83	90.2	82.1	94.9
No		9	9.8	5.1	17.9
**Who trained treatment staff on protocols for sick cows?**	48	83			
Include veterinarian		48	57.8	46.8	68.2
Dairy personnel only		35	42.2	31.9	53.2
**How often are protocols reviewed/revised?**	45	85			
Once to twice a year		56	65.9	55.0	75.3
Every few years		9	10.6	5.5	19.3
When a new product is added		20	23.5	15.6	33.9
**Who reviews/revises protocols?**	42	89			
Include veterinarian		65	73.0	62.7	81.4
Dairy personnel only		24	27.0	18.6	37.3

^1^ Others = product drug label, drug company material or sales rep, local/national meetings, state/county/university cooperative extension, websites, magazines/industry trade journals, food animal residue avoidance databank, and previous experience with the drug. ^2^ Therapeutic = disease specific treatment; ^3^ prophylaxis = vaccination, hoof trimming.

**Table 5 microorganisms-09-01507-t005:** Summary of responses to questions about antimicrobial drug (AMD) selection and tracking practices from responses to questionnaire on AMD use in adult cows on California dairies during 2019.

				95% Confidence Limits	
Antimicrobial Selection and Tracking Practice	Missing Data	*n*	Estimate (%)	Lower	Upper
**Do you keep a drug inventory log?**	6	125			
Yes		47	37.6	29.5	46.5
No		78	62.4	53.5	70.6
**Number of drug details recorded?** ^1^	0	131			
At least two		46	35.1	27.3	43.8
Only one		66	50.4	41.8	59.0
None		19	14.5	9.4	21.7
**How are AMD doses for cows estimated?**	9	122			
Veterinarian input		72	59.0	50.0	67.5
No veterinarian input		50	41.0	32.5	50.0
**How is AMD treatment duration determined?**	7	124			
Veterinarian input		94	75.8	67.4	82.6
No veterinarian input		30	24.2	17.4	32.6
**Factors that influence selection of a second AMD**	21	110			
Bacterial culture/veterinarian/protocol		90	81.8	73.3	88.0
Previous results		20	18.2	12.0	26.7
**Which AMD treatment information do you track/record?**	7	124			
Milk and meat withdrawal interval + others ^2^		84	67.7	58.9	75.5
Milk or meat withdrawal interval + others ^2^		14	11.3	6.8	18.3
No milk or meat withdrawal interval		26	21.0	14.6	29.1
**How do you track AMD treatments?**	2	129			
Computer ^3^		83	64.3	55.6	72.2
No computer		46	35.7	27.8	44.4
**How do you track AMD withdrawal period?**	5	126			
Computer + others ^3^		72	57.1	48.3	65.6
No computer		54	42.9	34.4	51.8

^1^ Details of drug-related information recorded by dairies farmers include name of drug, cost of drugs, quantity on hand, date of purchase, drug supplier/source, and drug expiration date. ^2^ Others include date of treatment, dose, and route. ^3^ Others include paper records kept in barn or office, markings on the animal, and white/chalk board or other temporary markings.

**Table 6 microorganisms-09-01507-t006:** Summary of responses to questions about veterinarian–client–patient relationship (VCPR) and disease diagnosis practices, from responses to questionnaire on antimicrobial drug (AMD) use in adult cows on California dairies during 2019.

				95% Confidence Limits	
Veterinarian–Client–Patient Relationship and Disease Diagnosis Practices	Missing Data	*n*	Estimate (%)	Lower	Upper
**Do you have a veterinarian–client–patient relationship?** ^1^	8	127 ^1^			
Yes		116	91.3	86.4	96.3
No		8	6.30	2.1	10.5
**Type of veterinarian involved with the VCPR**	12	119			
Local veterinarian/clinic		117	98.3	93.4	99.6
Consultant veterinarian		2	1.7	0.4	6.6
**Best description of VCPR**	15	116			
Written agreement		78	67.2	58.1	75.3
Verbal agreement		38	32.8	24.7	41.9
**How often does your veterinarian observe or discuss the health of your cows**	7	124			
Within a week		36	29.0	21.6	37.7
Within a month		62	50.0	41.2	58.8
More than month		6	4.9	2.2	10.5
As needed		20	16.1	10.6	23.8
**Have you submitted non-routine samples for infectious disease diagnosis in 2019**	4	127			
Yes		49	38.6	30.4	47.4
No		78	61.4	52.6	69.6
**Have you used on-farm diagnostic techniques to guide AMD treatment decisions**	3	128			
Yes		63	49.2	40.6	57.9
No		64	50.0	41.3	58.7
I do not know		1	0.8	0.1	5.5
**Have you used automated data collection systems for identifiying sick cows**	5	126			
Yes		19	15.1	9.7	22.5
No		107	84.9	77.4	90.2

^1^ Three surveys completed by veterinarians were excluded from this question.

**Table 7 microorganisms-09-01507-t007:** Summary of practices and perspectives from responses to questionnaire on antimicrobial drug (AMD) use in adult cows on California dairies during 2019.

				95% Confidence Limits	
Antimicrobial Stewardship	Missing Data	*n*	Estimate (%)	Lower	Upper
**Do you participate in an animal welfare audit program**	5	126			
Yes		108	85.7	77.6	95.8
No		18	14.3	9.1	21.7
**Type of animal welfare audit program**	5	126			
National program ^1^		107	84.9	77.5	90.2
Local program ^2^		1	0.8	0.1	5.6
None		18	14.3	9.1	21.7
**Do you participate in a dairy quality assurance program**	11	120			
Yes		55	45.8	37.0	54.9
No		65	54.2	45.1	63.0
**Are you familiarity with the FDA** ^3^ **term** **MIAD** ^4^	6	125			
Not sure/Not familiar		47	37.6	29.5	46.5
Aware that MIAD are available only via prescription		78	62.4	53.5	70.6
**Are you aware that MIAD require prescription, and are no longer sold OTC ^g^ since 2018**	5	126			
Yes		122	96.8	91.7	98.8
No		4	3.2	1.2	8.3
**Have you used OTC or prescription AMD on your dairy before January 2018**	6	125			
Both OTC and prescription AMD were used		78	62.4	53.5	70.6
Cows were only treated with prescription AMD		22	17.6	11.8	25.4
Cows were only treated with OTC AMD		4	3.2	1.2	8.3
Cows were not treated with OTC AMD		14	11.2	6.7	18.1
Cows were not treated with prescription AMD		1	0.8	0.1	5.6
**Have you made changes regarding previously available OTC AMD since January 2018**	14	117			
No changes made		61	52.1	43.0	61.2
Less AMD used		55	47.0	38.0	56.2
More AMD used		1	0.9	0.1	6.0
**Have you used or increased use of alternatives to AMD since January 2018**	5	126			
Yes		36	28.6	21.3	37.2
No		90	71.4	62.8	78.7
**Have you made changes to prevent disease outbreaks/spread since January 2018**	8	123			
Yes		35	28.5	21.1	37.2
No		88	71.5	62.8	78.9
**AMD drug cost since January 2018**	9	122			
Increased		16	13.1	8.1	20.5
Deceased		32	26.2	19.1	34.9
No change		74	60.7	51.6	69.0
**Farm animal health since January 2018**	12	119			
Better		51	42.9	34.2	52.0
Worse		5	4.2	1.7	9.8
No change		63	52.9	43.9	61.9

^1^ National programs = National Dairy FARM Program, Validus Dairy Animal Welfare Review Certification, Certified Humane^®^ Program. ^2^ Local programs = Dairy Farmers of America, Creamery, On-farm training, Cooperate extension. ^3^ FDA = U.S. Food and Drug Administration. ^4^ MIAD = medically important antimicrobial drugs. ^g^ OTC = over the counter.

**Table 8 microorganisms-09-01507-t008:** Summary of multiple factor analysis of conventional dairies showing seven identified components extracted from 84 variables collected from 113 responses to a questionnaire on antimicrobial drug (AMD) use in adult cows on California dairies during 2019.

Identified Components	Variation Proportion (%)	Component Variables	Correlation
**Good general practices**	18.4	Harvest colostrum from fresh cows to feed newborn calves	0.496
		Have a separate pen for recently calved cows	0.518
		Vaccinate adult cows for different diseases	0.410
**AMD usage information**	17.5	Sources info on AMDs used to treat cows	0.495
		Who decides which oral AMDs are purchased and stocked	0.482
		Who decides which AMDs are used to treat sick cows	0.401
**Mastitis management practices**	9.0	Mastitis: Basis for treatment decision	0.494
		Mastitis: Treat with intramammary and oral/injectable antibiotic	0.478
		Mastitis: Classes of first choice intramammary AMD infusion	0.401
**Metritis management practices**	11.6	Metritis: Basis for treatment decision	0.613
		Metritis: Treat with intrauterine, oral, and injectable AMDs	0.559
**Pneumonia management practices**	8.9	Pneumonia: Treatment bolus/injectable treatment	0.489
**AMD use stewardship practices**	7.0	Administration of appropriate AMD, dose, route, and duration	0.450
		Good record keeping on treatments and treatment dates	0.401
**Producer perceptions of antimicrobial resistance on dairies**	7.6	Current antibiotic use practices will make it harder to treat future infections	0.401
		Antibiotic use in livestock does not cause problem in humans	0.413
		Antibiotic use in livestock leads to bacterial infections in people	0.423

## Data Availability

This study was sponsored by the California Department of Food and Agriculture and is subject to California Food and Agriculture Code (FAC) Sections 14400 to 14408. FAC Section 14407 requires that data collected be kept confidential to prevent individual identification of a farm or business; as such, raw data from this study are not able to be shared.

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
