# Peer review of "2019 Survey of Antimicrobial Drug Use and Stewardship Practices in Adult Cows on California Dairies: Post Senate Bill 27"

_microorganisms, 2021, doi:10.3390/microorganisms9071507_

Round 1

Reviewer 1 Report

This manuscript is reporting results from a survey conducted on dairy herd in California in 2019. The survey was conducted 1 year after a regulation change about mandatory prescription of antimicrobials. The topic is of interest to the field. However, the manuscript is too long and the result section must be shortened a lot. There is no need to repeat in the text what can be visualized in the tables and figures. One or the other should be done, not both like the authors have done in many sections of the results. See specific comments below.

Line 20. A dot is missing.

Line 23. Define AMD at first appearance.

Lines 83-85. Please clarify the hypothesis. What king of impact? Positive?  negative? Magnitude? Please be more specific.

Line 104. Please explain what is a licensed grade A dairy in CA. Explain why these herds are representative of herds in CA.

Lines 114-115. This is repetitive.

Lines 162-163. Please specify the state mean.

Results section.

The result section must be shortened. Several results are presented in the text and in the tables. Do not repeat in the text what is presented in the tables.

All tables. Please put only 1 digit after dot. There is no need to have to digit. Also, the question column is not very clear. The category titles should be easier to distinguish from the answers (italic, bold, etc...). Finally, the n often do not add up to 131. It is not explained anywhere why there is so much variation for each question.

Table 1. Typo. Owner-manager.

Line 219. TS only should not be included in blanket dry-cow therapy as it is assumed that BDCT is including an antibiotic. Please put them separately.

Line 221. What is the difference between TS only here and line 219?

Lines 221-223. It is mutually exclusive. This indicates that they didn't understand the question... Should be excluded if the answer makes no sense. Looking at the questionnaire, there was no definition of blanket vs selective dry cow therapy. This could explain why the respondents struggle giving a clear answer.

Line 240. Put Staphylococcus in italics.

Line 447. VCPR. Define at first appearance.

Lines 615-627. This section could be expanded since it was one of your major objective and what the title and introduction lead the reader to expect.

Discussion section. Several countries have restrictions on category 1 antimicrobials (Highly critically important antimicrobials). Nothing is mentionned about this in this manuscript. What is the use of the category 1 antimicrobials in CA? Is it an  issue? There was no question in the questionnaire specifically about this. This is missing. Should be discussed.

Lines 776-778. Some producers are doing homemade solution using saline and some antibiotics or anti-inflammatory. This could be an explanation too.

Reviewer 2 Report

Dear authors,

Your survey is important and provides an interesting picture of antimicrobial use and stewardship practices in California dairy farms in 2019. The amount of data collected and presented is substantial. However, your analyses lack scientific rigor and some inconsistencies need to be addressed.

My main concerns are: (i) Your objectives and hypotheses are misleading. You did not compare before/after the new legislation (you present results from one survey, that was undertaken in 2019, and you do not compare your results to results from a study undertaken under the same conditions before SB-27). Therefore, you cannot indicate that you are evaluating “the impact” of the new legislation on practices. Your objectives and hypotheses need to be reformulated. (ii) Your statistics: you present many proportions, but your denominator is never the same. In your tables, you refer to 131 responses, but it seems that your denominator varies a lot, probably because of missing data. It will be important to clearly indicate the denominator for each proportion calculated, and to mention missing data. There are many inconsistencies between the numbers presented in tables and the same numbers in main text. Please verify and correct accordingly. Consider removing the standard error from tables (easily obtained from n and estimated proportion), but keep the 95% confidence interval. In addition, you do not exploit your data to their full potential. You present raw proportions, but your article would benefit from more advanced statistics (correlations, for example) and interpretations. (iii) The structure of your article should be reviewed. For example, you present results that were not described in materials and methods. Overall, I would suggest to prioritize the information obtained from your survey (main results versus results that are not relevant, for example because the answers are difficult to interpret or because the validity is questionable) and to place secondary results as supplementary material. It will help the reader to grasp important results from your study.

Round 2

Reviewer 1 Report

Thank you for considering my comments. Authors answered to the majority of them properly. However, the results section is still too long in this reviewer opinion. Some results are still repeated in the text and in the tables. It should be completely avoided. Do not repeat any results in the text that is already presented in the tables. Maybe some results should be move to the appendix. Also, adding missing data in the tables is not useful. The readers are able to do the math knowing that the total number of questionnaire was 131. I suggest to delete these additions. My question was why is there so much variation from one question to the other in the number of missing data. Should be discussed somewhere.

Reviewer 2 Report

Dear authors,

Thank you for your answers. The corrections that you applied improved the overall clarity of your article, and the reviewed version of the manuscript is improved compared to the first version. I maybe missed the information, but you may want to clarify why you collected a second time the answers to the same questionnaire (2018 and 2019). Especially with the hypothesis that practices would not have changed between 2018 and 2019. It is difficult to clearly understand the interest if you did not even expect any change. In addition, it seems curious that you had to “test the validity” of your questionnaire if you kept the same questions as the one administered in 2018. I would avoid the expression “validity” or “validity” because criteria are required in order to “validate” a questionnaire (see for example the chapter “Questionnaire design” in the textbook “Veterinary Epidemiology Research” for more details). You could say that the questionnaire was “pre-tested” in order to identify questions that were confusing or ambiguous.

Best regards.
